# Synthetic Approaches to Biologically Active C-2-Substituted Benzothiazoles

**DOI:** 10.3390/molecules27082598

**Published:** 2022-04-18

**Authors:** Bagrat A. Shainyan, Larisa V. Zhilitskaya, Nina O. Yarosh

**Affiliations:** A.E. Favorsky Irkutsk Institute of Chemistry, Siberian Branch of the Russian Academy of Sciences, 1 Favorsky Street, 664033 Irkutsk, Russia; lara_zhilitskaya@irioch.irk.ru (L.V.Z.); yarosh.nina@irioch.irk.ru (N.O.Y.)

**Keywords:** benzothiazole, synthesis, reactivity, biological activity

## Abstract

Numerous benzothiazole derivatives are used in organic synthesis, in various industrial and consumer products, and in drugs, with a wide spectrum of biological activity. As the properties of the benzothiazole moiety are strongly affected by the nature and position of substitutions, in this review, covering the literature from 2016, we focus on C-2-substituted benzothiazoles, including the methods of their synthesis, structural modification, reaction mechanisms, and possible pharmacological activity. The synthetic approaches to these heterocycles include both traditional multistep reactions and one-pot atom economy processes using green chemistry principles and easily available reagents. Special attention is paid to the methods of the thiazole ring closure and chemical modification by the introduction of pharmacophore groups.

## 1. Introduction

Benzothiazole and its numerous derivatives of electron-rich aromatic heterocycles with endocyclic sulfur and nitrogen atoms have attracted the ongoing interest of synthetic chemists due to their unique properties [1,2,3,4,5,6,7]. Recently, we have reviewed modern trends in the synthesis of biologically active and industrially important derivatives of 2-mercapto- and 2-aminobenzothiazoles [8,9]. The benzothiazole ring is the key motif of a wide range of biologically active compounds, including antitumor [7,10,11,12,13,14,15,16,17,18,19,20,21,22,23,24,25,26,27], antimicrobial [28,29,30,31,32,33,34,35,36], antiviral [37,38], antibacterial [16,24,34,37,39,40], antifungal [13,16,28,34,35,40,41,42], antiparasitic [32,43,44], antioxidant [19,45], antidiabetic [46], immunomodulating [47], and anti-inflammatory agents [48,49,50]. Some pharmacologically important C-2-substituted benzothiazole derivatives, such as antidiabetic Fortress, antitumor drugs Zopolrestat and GW 608-lys 38, and antiseptic Haletazol, have found application as commercially available drugs [3,51,52,53]. C-2-substituted benzothiazoles are also potential sensibilizers [54,55,56,57] and optically active materials [58,59,60,61,62,63,64,65,66,67,68,69,70,71,72,73,74]. With this in mind, the present review is devoted to the synthesis and practical application of various 2-substituted benzothiazoles, mainly covering the last five years. Nowadays, much attention is paid to minimizing the formation of toxic organic compounds by applying the methods of green chemistry. The effectiveness of different reactions can be increased by the use of nanocatalysts [75,76,77,78,79,80,81,82,83], silica- and nanosilica-based catalysts or oxidants [50,84,85,86,87,88], photocatalysts [89,90,91], solvent-free reactions [50,67,92,93,94,95,96,97], and the use of ionic liquids or ecologically friendly solvents, such as water or ethanol [98,99,100,101]. The effectiveness of reactions can be also increased by microwave [24,39,50,102] or visible light assistance [17,18,41,91,103,104]. However, along with one-pot atom economy reactions, multistep processes are still widely used for the synthesis of C-2-substituted benzothiazoles. Nowadays, in the design of new drugs, the concept of molecular hybridization is actively used. This concept means combining two or more moieties of different biologically active compounds, each of which is known to possess pharmacological activity, in new hybrid molecules, resulting in the enhancement of biological effects and overcoming drug resistance [10,11,12,17,18,19,22,23,32,33,34,46,105,106]. Below, the syntheses of the C-2-substituted benzothiazoles are classified according to the methods of their formation and functionalization.

## 2. Intramolecular Formation of the C-2-Substituted Benzothiazole Ring

Benzothiazoles **1a**–**y** with alkyl, aryl and hetaryl substituents in position 2 of the ring were prepared in moderate to good yields by a metal-free atom-economic procedure [107]. The cascade process and the R_3_ group transfer were initiated by di(*t*-butyl)peroxide (DTBP) in fluorobenzene. The reaction started with the homolytic fission of DTBP upon heating to give *t*-butoxy radical, which suffered β-scission to give methyl radical. The proposed mechanism is presented in Figure 1. 

The copper NHC complex-catalyzed intramolecular S-arylation of various 2-halogenothioanilides was investigated as a route to 2-arylbenzothiazoles **2a**–**f** [108] (Figure 2). Good yields were obtained both for electron donor and electron acceptor substituents in the aryl rings. The mechanism, including two-electron Cu(I)/Cu(III) catalytic cycles with the intramolecular cyclization of 2-halogenothioanilides to 2-arylbenzothiazoles, was proposed.

## 3. Intermolecular Formation of the C-2-Substituted Benzothiazole Ring

There are many protocols for the design of a benzothiazole ring based on the transition metal catalysis or metal-free syntheses using one-pot processes carried out in the absence of a solvent or in “green” solvents. Thus, the cascade radical cyclization of *ortho*-isocyanoaryl thioethers with organoboric acids promoted by Mn(acac)_3_, FeCl_2_, CuCl_2_ or benzoic peroxyanhydride (BPO) led to various C-2-substituted benzothiazoles **3a**–**r** in 47–89% yield (Figure 3); the reaction successfully occurred in toluene, fluorobenzene, or ether [109]. The stepwise radical mechanism is similar to that in Figure 1. 

The alternative visible light-induced, metal-free and oxidant-free cyclization of *ortho*-isocyanoaryl thioethers with ethers provides an efficient route to benzothiazoles functionalized with ether groups **4****a**–**w** (Figure 4). As a photocatalyst, 1,2,3,5-tetrakis-(carbazol-9-yl)-4,6-dicyanobenzene (4CzIPN) was used [41]. A similar stepwise radical mechanism was triggered by the excitation of the photocatalyst to 4CzIPN and the single-electron transfer from the ether on 4CzIPN to give α-oxy radical, which reacts with isocyanaryl to form the imidoyl radical. Finally, the intermolecular cyclization of the latter resulted in the formation of the target product and the elimination of the methyl radical (Figure 4). 

The synthesis of 2-substituted benzothiazoles **5a**–**z** from *o*-iodoarylisothiocyanates and a series of methylene active compounds mostly in quantitative yield has been reported [110]. The reaction is transition metal-free and proceeds at room temperature in the presence of sodium hydride by the formation of an intramolecular C–S bond. The authors proposed the S*_RN_*1 mechanism with the formation of radical intermediates (Figure 5). Sodium hydride reacts with the active methylene compound to give carbanion, which adds to the isothiocyanate group to form the thioamide intermediate (**A**). Under alkaline conditions, the latter is transformed to the conjugate base (**B**), in which a single electron is transferred to the aryl group with the formation of the radical-anion intermediate (**C**). The latter expels the iodide ion, resulting in biradical intermediates (**D**) which, in turn, undergo intramolecular recombination to the target products (Figure 5). 

Condensation of substituted anilines with benzoyl chlorides with subsequent thionylation with the Lawesson reagent (2,4-bis(4-anisyl)-1,3,2,4-dithiaphosphetane-2,4-disulfide) and Yacobsen cyclization of thioanilides under the action of alkaline solution of K_3_Fe(CN)_6_ affords 4-nitrophenyl benzothiazoles **6a**–**f**. The latter were reduced with SnCl_2_ to the corresponding 4-aminophenyl benzothiazoles **7a**–**f** in 75–80% yield (Figure 6) [12]. The condensation of compounds **7a**–**f** with aromatic ethynyl ketones in ethanol affords arylaminobenzothiazole-arylpropenones hybrids **8****a**–**r** in high yield. The authors demonstrated cytotoxic activity of the obtained products. 

Fluorinated or perfluoroalkylated 2-methylbenzothiazoles **9a**–**h** and **10a**–**h** were synthesized from fluoro- or perfluoroalkylanilines in three steps: acylation of the amino group, transformation of the carbonyl group to thiocarbonyl, and catalyzed cyclization (Yacobsen reaction). The obtained 2-methylbenzothiazoles **9a**–**h** gave benzothiazolium tosylates **10a**–**h** by heating with methyl tosylate (Figure 7) [43]. 

Tosylate salts **10a**–**h** have been used as building blocks for the design of fluorinated rhodacyanines **11a**–**q**, which demonstrated high antileishmanial activity (Figure 8) [43]. 

The reaction of anilines with sulfinylbis[(2,4-dihydroxyphenyl)methanethione] gives benzothiazoles **12a**–**c** a 2,4-dihydroxyphenyl substituent in position 2 of the benzothiazole ring (Figure 9). The reaction starts with electrophilic substitution and the HF or HCl elimination from the formed thioamide. The perfluorinated product has shown notable activity against human cancer cells [13].

A series of new “head-to-head” aniline-based derivatives of bis-benzothiazole were obtained and their antiproliferative activity was assessed [14]. In the presence of Br_2_, benzidine reacts with potassium thiocyanate via cyclization to bis(benzothiazole)diamine. Its hydrolysis with KOH leads to the key intermediate, 3,3’-bis(mercapto)benzidine. The latter reacts with *p*-substituted benzaldehydes to give bis-substituted benzothiazoles **13a**–**j** (Figure 10). The products with electron-donor substituents in the benzene ring are less toxic and more effective.

DMSO acts both as the solvent and the oxidant in the metal-free ecologically safe synthesis of C-2-substituted benzothiazoles **14a**–**g** and naphtho [2,1-*d*]thiazoles **15a**–**z** from *N*-substituted arylamines and elemental sulfur (Figure 11) [111]. The advantages of the method are the use of easily accessible anilines, a variety of 1 and 2-naphthylamines and 2-anthranylamine, and tolerance to a wide range of functional groups. 1,3 and 1,4-bisnaphtho [2,1-*d*]thiazoles linked by the benzene bridge have also been synthesized. The electron-donating groups in the aniline fragment notably increase the yield of the target products. The proposed mechanism is shown in Figure 11, using the example of naphthylamine. First, amine is oxidized by DMSO to imine (**A**). The electrophilic attack of elemental sulfur S_n_ to the *ortho*-position of imine (**A**) gives intermediate (**B**). The elimination of sulfur S_n-1_ and the proton results in the imine thiolate (**C**), which undergoes nucleophilic intramolecular cyclization to thiazoline (**D**). Finally, oxidative aromatization of the latter gives rise to the target annelated products **15**.

Most reactions of intermolecular formation of the C-2-substituted benzothiazole ring are based on the use of readily accessible 2-aminothiophenols and green chemistry principles. An example is the reaction of direct oxidative condensation of aminothiophenols and aliphatic, heterocyclic or aromatic alcohols to benzothiazoles **16a**–**m** with different substituents upon irradiation with visible light in the presence of a photocatalyst (Figure 12) [91]. The process is scalable and economic; the yield of the products depends on the electronic and steric effects of the alcohol molecule. The reaction mechanism includes the oxidation of alcohols to aldehydes, the condensation of the latter with *ortho*-aminophenols to imine/benzothiazolines, and their oxidation to 2-substituted benzothiazoles. 

Another example is the green synthesis of benzothiazoles **17a**–**t** by the condensation of 2-aminothiophenol with various aldehydes in the presence of heterogeneous catalysts. As such, SnP_2_O_7_ prepared from monoammonium phosphate and SnCl_2_ solution, or Sm(NO_3_)_3_·6H_2_O applied on nanosized silica gel, were used. As solvents, ethanol or methanol were employed [85] (Figure 13, upper reaction). The catalysts can be recycled five times without notable loss of the catalytic activity. Benzaldehydes with electron acceptor or electron donor groups, as well as heterocyclic aldehydes, readily entered the reaction with 2-aminothiophenol (yields: 85–96%); lower yields (68–73%) were obtained for aliphatic aldehydes. However, with microwave assistance, the yield of the reaction of 2-aminothiophenol with aliphatic aldehydes may reach 98%. The reaction was carried out without solvent in the presence of charcoal and silica gel (Figure 13, bottom reaction) [50]. Microwave assistance in the presence of catalytic amounts of Amberlite IR-120 resin also allowed the authors to obtain a large series of aryl- and hetarylbenzothiazoles **18a**–**g** containing different functional groups from aldehydes and 2-aminothiophenol [24]. 

In other ecologically friendly syntheses of C-2-substituted benzothiazoles from aminothiophenols, cheap water-soluble urea nitrate [35], ionic liquid with the sulfonate anion group playing the role of the heterogeneous catalyst and the solvent (BAIL GEL) [100], or a biocatalyst in the form of a natural carrier of calcined limpet shells coated with ZnCl_2_ were used [112].

A simple and efficient synthesis of 2-alkylbenzothiazoles **19a**–**g** was performed by a two-step reaction including the condensation of 2-aminothiophenol with aliphatic aldehydes in the presence of molecular sieves 4Å followed by the oxidation of the formed 2-alkyl-2,3-dihydrobenzo[*d*]thiazoles with pyridinium chlorochromate (PCC) on silica gel (Figure 14) [84].

Distinct from aldehydes, ketones react with 2-aminothiophenol via their active methylene group, as proven by the carbonyl group remaining intact in the products. Thus, a series of aromatic 2-acylbenzothiazoles **20a**–**k** was obtained from 2-aminothiophenol, in addition to aromatic or heteroaromatic ketones by reflux in ethanol with CuBr_2_ as the oxidant (Figure 15) [101]. Apparently, the reaction proceeds with N-nucleophilic substitution in α-bromoketone generated from the ketone and CuBr_2_. The formed α-aminoketone is further brominated by CuBr_2_ and cyclized by the nucleophilic attack of the thiol group on the α-carbon atom with the elimination of HBr and the closing of the ring, as shown in Figure 15. In the final step, dehydrogenation with a reduction of CuBr_2_ to CuBr gives the target 2-acylbenzothiazoles **20a**–**k**.

For the synthesis of new benzothiazole-based hemicyanine sensitizers for solar cells, the ring closure was performed by the reaction of 2-aminothiophenol with isopropyl methyl ketone in the presence of acetic anhydride. Then, 2-methylbenzothiazole formed in a practically quantitative yield reacted with 1,2-oxathiane 2,2-dioxide to give the corresponding sulfonates and, finally, by the reaction with dimethylaminobenzaldehyde or 3,4-dihydroxycyclobut-3-ene-1,2-dione, new sensitizers **21** and **22** were formed (Figure 16) [56,57].

Several groups have developed the synthesis of C-2-substituted benzothiazoles **23a**–**c** from 2-aminothiophenols and β-diketones by the use of effective, recycled, cheap and ecologically safe catalysts, such as the montmorillonite clay KSF [113], long-chain ionic liquids [114], sodium dichloroiodate [115], or the Zr-based organometallic catalyst MOF-808 [116]. The mechanism given in Figure 17 is an example of condensation with the participation of montmorillonite clay [113]. The reaction includes keto-enol tautomerization, the formation of enaminoketone, its cyclization, and the elimination of the enolate. The catalyst is easily separated by simple filtration. 

The reaction of the acylation of 2-aminothiophenol with acetic acid by the action of direct concentrated solar radiation on heating in the presence of choline chloride has been studied. The yield of product **24a** was 60% (Figure 18, upper route) [117]. The authors note the chemoselectivity of the process of intramolecular acylation. Choline chloride forms hydrogen bonds with the carbonyl oxygen, thus activating the reagent; moreover, it acts as a phase-transfer catalyst and activates the aniline moiety, facilitating the nucleophilic attack and the formation of the intermediate N-acylated product. The method is a good example of green synthesis, as it is metal-free, oxidant-free, and uses choline chloride, which is an inexpensive, biodegradable and recycled catalyst which can be used in water medium.

A similar approach to 2-methylbenzothiazole **24a** from aminothiophenol and malonic acid was described [118]. The method is simple, scalable, and gives only small amounts of by-products (Figure 18, bottom route).

The yields of compound **24a** up to 95% were obtained when using such catalysts as nanoporous TiO_2_ modified with bis-3-(trimethoxysilylpropyl)ammonium hydrosulfate (TiO_2_-[bip]-NH_2_HSO_4_) [95], a polymer-based solid acidic catalyst [PVP-SO_3_H]HSO_4_ [96], or a nanocatalyst on mesoporous silica containing bridge groups of *N*-sulfonic acid (SA-PMO) [97]. All reactions were carried out under mild conditions and without solvent.

A simple one-pot synthesis of 2-substituted benzothiazoles **25a**–**k** by the reaction of acid chlorides or anhydrides with 2-aminothiophenol in the presence of a basic heterogeneous catalyst KF·Al_2_O_3_ was proposed (Figure 19). The reaction proceeded under mild conditions in high yields, and the catalyst did not lose its activity after 10 times of recycling. No by-products were detected, and the target products were isolated by simple filtration [119].

A convenient route to 2-organyl benzothiazoles **26a**–**k** from 2-aminothiophenols and the derivatives of dimethylformamide in moderate to high yields without the use of toxic solvents has been reported [92]. The reaction performed in the presence of imidazolium chloride was shown to be sensitive to temperature: lowering the temperature by 20 °C decreased the yield by six times. The authors assume that the reaction was initiated by the activation of DMF derivatives with imidazolium chloride leading to the intermediate tetrahedral compound (**A**). Its decomposition resulted in the formation of the intermediate protonated *N*-acylimidazole (**B**), which launched a series of transformations of the substrate resulting in cyclization (Figure 20).

Non-catalyzed cyclocondensation of 2-aminothiophenol with 4-methylbenzaldehyde in DMSO at 190 °C affords 2-(4-tolyl)benzothiazole. The latter undergoes a sequence of transformations leading to dendrimers with terminal benzothiazole groups **27a**–**c** (Figure 21). Similar reactions were performed with 4-methylcinnamic acid. Photophysical investigation of the obtained dendrimers showed a possibility of their use as additives to sensitized dyes in solar cells [54]. 

Now, let us turn to the light-induced syntheses of C-2-substituted benzothiazoles. The method of the synthesis of 2-organylbenzothiazoles **28a**–**s** was developed based on the photooxidative cross-coupling of 2-aminothiophenols with α-oxocarboxylic acids under the action of blue UV irradiation in the presence of H_2_O_2_ (Figure 22). The key step of the radical mechanism of the reaction is the formation of the donor acceptor complex between the reagents. Subsequent decarboxylation and intramolecular cyclization of the intermediate adducts afford the target products. α-Ketoacids and 2-aminothiophenols with various functional groups react readily at room temperature in moderate to good yields without the use of photooxidative or metal-based catalysts [103].

Visible light-induced cascade radical cyclization was performed for the synthesis of benzothiazoles possessing CF_2_/CF_3_ substituents in the 2-position, **29a**–**k** and **30a**–**k,** in good yield (Figure 23). The use of Na_2_CO_3_ as a reducing agent facilitated mild fluoroalkylation [90].

The visible light-induced reaction of 2-aminothiophenols with aldehydes was proposed as an economic and safe route to a wide series of benzothiazoles, **31**, affording the target products in good yields in the absence of transition metal catalysts or other additives (Figure 24) [104]. The authors proposed a radical mechanism via diaryldisulfide intermediates.

A series of benzothiazolamides, **32a**–**l**, possessing antimicrobial and antifungal activity was prepared in high yields via the cyclocondensation of 2-aminothiophenol with diethyl oxalate, the hydrolysis of the formed ethyl benzothiazole-2-carboxylate, and amidation with the amides of 4-nitrophenylalanine in the presence of HATU (hexafluorophosphate azabenzotriazole tetramethyl uronium) and DIPEA (diisopropylethylamine) in DMF (Figure 25) [28].

Cyclization of 2-aminothiophenol with acetyl chloride affords 2-methylaminobenzothiazole, which, when treated with bromoacetic acid, gives 3-carboxymethyl-2-methylbenzothiazolium bromide. The latter enters condensation with aldehydes in acetonitrile in the presence of piperidine as a base to give new chromophores **33a**–**e** containing the benzothiazole moiety and alkyl groups of different chain lengths (Figure 26). The investigation of photoelectric properties showed that the efficiency of the power transformation for all sensitizers **33a**–**e** increased with the length of the carbon chain [55].

2-Alkyl- and arylsubstituted benzothiazoles **34a**–**o** were synthesized by the solvent-free and metal-catalyst-free reaction of 2-aminothiophenols and N-organylthioamides in the presence of CBr_4_ (Figure 27). The reaction includes the activation of thioamide by the formation of the intermediate with the S–Br bond between the thioamide sulfur atom and CBr_4_. The activated thioamide molecule attacks aminothiophenol, and the reaction is completed by intramolecular cyclization and the formation of the target products and *N*-methylaniline, and the regeneration of the catalyst from H_2_S Br–CBr_3_. The yields for the aliphatic derivatives were 68–93%; for aromatic, 62–81% [93].

Disulfides can also be used as starting materials for the synthesis of C-2-substituted benzothiazoles. Thus, 2-alkyl and 2-aryl(hetaryl)benzothiazoles **35a**–**k** have been prepared by the oxidative coupling of (2-aminoaryl)disulfides and primary alcohols in the presence of initiator DTBP (Figure 28) [120]. The yields decreased with the steric volume of substituent R_2_ in the molecule of the alcohol. The highest yields were obtained for ethanol and benzyl alcohol. No reaction occurred with methanol or isopropanol. The process was initiated by the decomposition of DTBP on heating to *t-BuO* radicals, which oxidized the alcohol molecule. The stability of the formed radical plays a decisive role in, e.g., methanol forming an unstable primary radical. On the other hand, only primary alcohols can be used because two hydrogens in the α-position are necessary for radical oxidation.

Ecologically friendly, NaSH-promoted condensation of bis(2-aminophenyl)disulfides and aryl- and hetaryl aldehydes in polyethylene glycol with low-energy microwave assistance allowed to obtain 2-substituted benzothiazoles **36a**–**q** in good yield (Figure 29) [39]. The method is applicable to benzaldehydes with both electron donor and electron acceptor groups. The presence of NaSH facilitates the fast reduction of disulfides to aminothiophenols. The latter react with benzaldehydes affording the corresponding Schiff bases. Intramolecular oxidative cyclization accomplishes this process.

α-Ketoacids react with 2,2’-disulfanediyldianilines in the presence of Na_2_S_2_O_5_ via condensation with the amino groups and subsequent cyclization by the nucleophilic addition of sulfur to the C=N bond (Figure 30) [121]. The intermediate disulfides (**A**) suffer the S–S bond splitting and decarboxylation finally affords C-2-substituted benzothiazoles **37a**–**p** in moderate to excellent yields. The highest yields in the experiment were obtained for electron-withdrawing substituents in the aromatic ring of α-ketoacid. The reaction is metal-free and proceeds with the evolution of ecologically safe CO_2_. The presence of Na_2_S_2_O_5_ is required for complete conversion and for obtaining maximal yields of the target products. 

Very recently, the reaction of *ortho-haloanilides* with alkali metal sulfides was reported [122]. The reaction proceeds upon heating in DMF in the presence of heterogeneous catalyst MCM-41-NHC-CuI via the CuI-catalyzed substitution of halogen by sulfur, and cyclization with dehydration and regeneration of the catalyst (Figure 31). A series of C-2-substituted benzothiazoles **38** were obtained in good yields.

C-2-substituted benzothiazoles can also be prepared by different one-pot multicomponent reactions. Thus, the effective three-component reaction of redox cyclization allowed the authors to obtain a series of 2-arylbenzothiazoles **39** [123,124]. The reaction was easy to handle, catalyzed by cheap copper acetate, tolerated a wide range of functional groups, was scalable, and used readily available reagents: haloanilines, stable non-toxic arylacetic acids or benzyl chlorides, and elemental sulfur (Figure 32). The yields varied from good to excellent. The key step both in the reaction with arylacetic acids and with benzyl chlorides is the copper-catalyzed formation of diarylsulfides. 

An effective and ecologically friendly methodology has been described for the synthesis of C-2-substituted benzothiazoles **40a**–**l** and **41a**–**c** (Figure 33) [87]. The one-pot three-component reaction of 2-iodoaniline, aryl- or hetaryl aldehydes and thiourea was catalyzed by ferromagnetic catalyst Cu(0)–Fe_3_O_4_@SiO_2_/NH_2_cel and was carried out with water as the solvent. The catalyst was easily retrieved with a magnet. A large number of products were obtained in good yields, and the electronic effects in the substituents did not affect the course of the reaction.

The alternative metal-free reaction of anilines, elemental sulfur and ethers in the presence of TBHP and KI gives rise to 2-organylbenzothiazoles **42a**–**r** (Figure 34) [125]. The nature and position of the substituents in the aniline moiety have no substantial effect on the yield of the target products. The reaction with cyclic ethers proceeds with ring opening leading to heterocyclic alcohols **43a**–**c** in good yields. 

The cyclization of anilines is assumed to be initiated by the selective splitting of the C(sp^3^)–H bond in ethers in the presence of TBHP and KI. As a rule, the first step of reactions of this type is the formation of a radical, (here, *t*-BuO·). The latter is formed by the reaction of TBHP with KI.

Similar one-pot reactions leading to 2-hetarylbenzothiazoles from anilines, elemental sulfur and 2-methylquinolines or benzaldehydes have been described [126,127]. 

A three-component reaction of 2-aminothiophenols, oxalyl chloride and thiols in the presence of *n*-tetrabutylammonium iodide (TBAI) allowed the authors to obtain a wide series of *S*-alkyl- and arylbenzothiazol-2-carbothioates **44** (Figure 35) [30]. It was assumed that TBAI reacted with thiol to give thiolate ion, which attacked oxalyl chloride with the formation of the thioether intermediate entering TBAI-assisted condensation with 2-aminothiophenol to give the target products in 56–80% yields. The investigation of biological activity showed antimicrobial activity and low toxicity of the products. 

The metal-free assembly of C-2-substituted benzothiazoles **45a**–**g**, **46a**–**f** or **47a**–**l** based on the reaction of arylamines, elemental sulfur and styrenes or arylacetylenes in N-methylpyrrolidin-2-one (NMP) has been reported (Figure 36) [128]. The C–S bond was formed by direct thiylation of the C–H bond in aromatic amine with elemental sulfur, acting both as the source of sulfur and the oxidant. The addition of NH_4_I increased the yield, which was also affected by the nature and position of substituents in the phenyl ring. A possible mechanism for the formation of the C-2-substituted benzothiazoles is given in Figure 36, using the example of aniline with sulfur and styrene. Aniline reacts with sulfur to give adduct (**A**), which further reacts with styrene to give polysulfide (**B**). The latter adds another aniline molecule leading to thioamide (**C**). The S–S bond in the latter is split to form thioamide (**D**) which, after oxidative cyclization, affords the final product.

Later, the strategy of a highly atom economical Cu(II)-catalyzed assembly of benzothiazoles from 2-iodoanilines, alkenes and elemental sulfur—avoiding the use of ecologically undesirable thiophenols—was developed by another group [129]. 

## 4. Synthesis of C-2-Substituted Benzothiazoles via the Introduction of Substituents at the 2-Position 

A particular class of reactions is the functionalization of the already existing benzothiazole motif at the 2-position. This approach has already led to the synthesis of a large number of compounds including those possessing different pharmacological activity. For example, benzothiazoles are alkylated with acetonitrile at the 2-position in the presence of lithium *t*-butoxide and dioxane as a cosolvent to give 2-methylbenzothiazoles **48a**–**e** (Figure 37) [130].

A simple approach to 2-arylbenzothiazoles **49a**–**i** based on the coupling reaction between benzothiazole and arylsulfamates was proposed [131]. The reaction proceeds in the presence of a catalyst and cocatalyst with nickel bromide and 1,10-phenanthroline monohydrate (Figure 38).

A sequence of reactions including the acylation of benzothiazole and the amidoalkylation of indole at the 3-position with N-acylbenzothiazolium intermediate and oxidation of the formed products **50a**–**e** with *o*-chloroanil leading to benzocamalexin **51** (Figure 39) [132,133]. The latter is the benzo-analogue of the natural plant-produced antimicrobial substance phytoalexin inhibiting the growth of parasites. The method is advantageous over other methods of heteroaromatic ring coupling, as it does not require expensive catalysts of air- and moisture-sensitive organometallic reagents, results in high yields, and is scalable. 

The chemoselective alkylation/arylation of benzothiazoles with aldehydes and benzyl alcohols in the presence of a heterogeneous nanocomposite catalyst and oxidant with graphene oxide–Fe_3_O_4_ in polyethylene glycol (Figure 40) affords 2-alkyl(aryl)-substituted benzothiazole derivatives **52a**–**u** and **53a**–**g** in moderate to excellent yields [134]. The advantages of the method are the absence of noble metals, toxic solvents, easy product isolation, and the possibility of reusing the catalyst without the loss of catalytic activity. The reaction proceeds with the thiazole ring opening and the condensation of the formed aminothiophenol with aldehyde. Then, the formed imine (**A**) undergoes intramolecular cyclization with the formation of 2-substituted thiazoline (**B**) and aromatization of the latter by the action of oxidant DIAD (diisopropyl azodicarboxylate). 

A practical green synthesis of 6-substituted 2-(2-hydroxy(methoxy)phenyl)benzothiazoles **54a**–**f**, including mesylate salts **55a**–**f**, was elaborated (Figure 41) [15]. The reaction was catalyst-free and used the ecologically safe and cheap solvents of glycerol and acetic acid. The optimization of the reaction conditions, solvents, and the reagents allowed the authors to carry out the reaction with compounds with hydrolytically unstable substituents. The relationship between the structure and biological activity for new compounds was studied, such as 2-hydroxyphenyl- and 2-methoxyphenylbenzothiazole with different substituents in the C-6 position of the benzothiazole fragment. The presence of the nitro or cyano group in the C-6 position of the benzothiazole ring was found to increase the antiproliferative activity. The replacement of the cationic amidine fragment in the C-6 position by the ammonium group led to the increase in antitumor activity against other types of tumor cells. The presence of a hydroxy group in the 2-aryl fragment of 2-arylbenzothiazole molecule considerably improved the antitumor selectivity without affecting the surrounding tissues. 

Various acyl groups were introduced in benzothiazoles in the presence of a Fe(II) triflate catalyst by the reaction of benzothiazole and its derivatives with cyclobutanone oximes (Figure 42) [135]. A wide spectrum of alkylbenzothiazoloarylketones **56a**–**k** was synthesized with a good selectivity and tolerance to the functional groups. The proposed method was an alternative to the conventional Friedel–Crafts acylation, allowing the authors to prepare new compounds inaccessible by other methods. The mechanism included several steps: Fe(II)→Fe(III)-induced SET-reduction of cyclobutanone oximes leading to iminyl radical (**A**) и Fe(III); ring opening in (**A**) to form the highly reactive cyanoalkyl radical (**B**); the capture of CO to give radical (**C**); and the addition to benzothiazole resulting in the radical (**D**). The oxidation of the latter by Fe(III) with subsequent deprotonation with a base gives alkylhetarylketones **56a**–**k**.

## 5. Modification of Substituents in the C-2 Position of Benzothiazoles

The modification of substituents in the C-2 position is a widely used reaction; some examples are considered below. The condensation of N-benzyl-2-methylbenzothiazolium bromide prepared by the alkylation of 2-methylbenzothiazole with benzyl bromide and N-ethylcarbazole dialdehyde gives rise to the formation of the carbazole–benzothiazole hybrid fluorescent probe **57** (Figure 43) [106]. This fluorophore showed a quick response, in addition to high selectivity and sensitivity in the detection of SO_2_. Moreover, good biocompatibility and a precise localization in the mitochondria were found. 

A large series of potentially biologically active drugs, in particular, antitumor agents, based on benzothiazol-2-ylacetonitrile (**BTA**) has been described [10,11,16]. Below, some examples of the use of this synthon and the products thereof are given. In the synthesized hybrids, the benzothiazole fragment has different substituted heterocyclic rings in the C-2 position, such as thiazole, thiazinane, thiophene, pyrrole, thienopyrimidine, indole, furan, pyridine, chromene, quinoline, triazoloquinoline, triazepinoquinoline, etc. The pyridine or furan hybrids **58** or **59** are formed by the reaction of benzothiazol-2-ylacetonitrile containing an active methylene group with 2-(2,4-dimethoxybenzylidene)malononitrile or ethyl-2-chloro-3-oxobutanoate (Figure 44) [10]. 

The reaction of BTA with carbon disulfide gives ketene acetal, which reacts with α-chloroethyl acetate resulting in thiophenebenzothiazole **60**. Hydrazinolysis of the latter and condensation of the hydrazide with phthalic or acetic anhydride in the presence of acetic acid results in the corresponding amides **61** and **62** in good yields. The reaction of **BTA** with phenylisothiocyanate and phenacyl bromide affords the corresponding thiophene derivative **63** (Figure 45) [10]. Compounds **61** and **62** have shown high antitumor activity to different cell lines.

A similar two-step approach led to the thiazole-pyrazole **64** or -thiophene **65** hybrids (Figure 46) [10]. 

Compound **65** was further functionalized by the reaction of cyclocondensation with formic acid, chloroacetyl chloride, ethyl cyanoacetate, or ethylenediamine to give benzothiazole thienopyrimidine **66a**–**c** or the imidazoline derivative **67** (Figure 47) [11]. The latter compound, similar to compounds **61** and **62** above, has shown high antitumor activity to different cell lines.

The cyclization of compound **65** with Meldrum acid resulted in the formation of the tricyclic system **68** (Figure 48) [11].

The polyheterocyclic compound **69** containing a tetrazole ring was obtained by the treatment of product **60** (Figure 45) with triethyl formate and heating in acetic acid in the presence of sodium azide (Figure 49) [11]. 

The nucleophilic addition of the amino group of compound **60** to the cyano group of 2-(4-(4-chlorophenyl)thiazol-2-yl)acetonitrile with subsequent intramolecular cyclization and the elimination of ethanol leads to the formation of compound **70** with the thienopyrimidinone ring (Figure 50) [11]. 

The iminoquinoline derivative **71** was synthesized by the Knoevenagel reaction using bromosalicyl aldehyde as the carbonyl component and benzothiazol-2-yl acetonitrile, followed by intramolecular cyclization and reflux with hydrazine hydrate in ethanol (Figure 51) [10].

The cascade multicomponent reaction of product **71** with *p*-chlorobenzaldehyde and benzothiazol-2-ylacetonitrile in dioxane led to the formation of the triazepine derivative **72** (Figure 52) [11]. 

A series of biologically active compounds was obtained from 2-[3(4)-aminophenyl]benzothiazoles **73** or **74** [17,18,20,29]. Thus, the reaction of (3-aminophenyl)benzothiazole **73** with ethyl acetylacetonate with the subsequent formation of the pyrazole ring by the reaction with hydrazines afforded 2-benzothiazolyl pyrazole derivatives containing hydrazone spacers **75a,b** (Figure 53) [17].

Condensation of the isomeric 4-aminophenylbenzothiazole **74** with aromatic aldehydes or ketones in glacial acetic acid or in the presence of conc. H_2_SO_4_ leads to benzothiazoles with the azomethine bonds **76a**–**p** and **76q**–**s** (Figure 54) [18]. These Schiff bases show anticancer activity and compounds possessing dihydroxy groups with very high inhibitive activity.

With chloroacetyl chloride, compound **74** forms 2-substituted benzothiazole with chloroacetamide group **77** which, upon the reaction with substituted piperazines, gives 2-aryl benzothiazole derivatives **78a**–**o** possessing anticancer activity. The reaction of **74** with propargyl bromide followed by cyclization of arylazides to the triple bond gives products with the 1,2,3-triazole motif **79a**–**k** in 67–91% yields (Figure 55) [20]. 

With primary and secondary amines, compound **77** reacts with the formation of a large library of heterocyclic benzothiazole derivatives **80a**–**m** and **81a**–**o**, for which anticancer activity has been evaluated (Figure 56) [21]. 

The synthesis of azo-linked-substituted benzothiazoles **82** and **83** in good yield by the diazotation of 2-(5’-amino-2’-hydroxyphenyl)benzothiazole was reported [32]. Diazotation was performed under the usual conditions with subsequent treatment with N,N-dibutyl-4-phenylthiazole-2-amine or 3-(diethylamino)phenol in acidic medium upon cooling (Figure 57). The antibacterial activity of the obtained products was investigated.

Using the reaction of the diastereoselective ketene-imine cycloaddition, sixteen new benzothiazole β-lactam conjugates have been synthesized [33]. The reaction was performed by the treatment of (benzothiazol-2-yl)phenols with bromoacetic acid in DMF in the presence of solid K_2_CO_3_. The subsequent reaction of the obtained oxyacetic acids with the Schiff bases in the presence of tosyl chloride gave the target *cis-*β-lactams **84a**–**p** in yields from 60 to 90% (Figure 58). The obtained hybrids showed good antimicrobial and antimalarial activity. The presence of the nitrophenyl group at the C-4 atom of the β-lactam ring, or anisyl, tolyl, or naphthyl groups on the N-1 atom of the β-lactam ring enhances the antimicrobial activity. 

The introduction of 2-(4-hydroxyphenyl)benzothiazole in the reaction with propargyl bromide in the presence of a base affords 2-(4-propargyloxyphenyl)benzothiazole, which enters cycloaddition reactions with various azides in the presence of copper fluorapatite, leading to benzothiazole–triazole hybrid molecules **85a**–**t** (Figure 59) [22]. 

As mentioned above, a number of hydroxyl-derivatives of benzothiazole demonstrate fluorescent properties. For example, the synthesis of the benzothiazole-based water-soluble biochemosensor **86** used for the detection of intracell zinc and aluminum ions has been described [64]. For this, 3-(benzo[*d*]thiazol-2-yl)-2-hydroxy-5-methylbenzaldehyde is prepared by successive treatment of hydroxymethylphenylbenzothiazole with trifluoroacetic acid and diaminomalononitrile in the presence of catalytic amounts of acetic acid in dry ethanol (Figure 60).

Another green and efficient approach to luminophores is mechanochemical, solvent-free synthesis [65]. A mixture of 2-(2-hydroxyphenyl)benzothiazole, hexamethylenetetramine, trifluoroacetic acid and silica gel was thoroughly grinded for 3 h. The obtained product was purified by chromatography and grinded with benzophenone hydrazone for 0.5 h. The synthesized dye **87** (Figure 61) can be used for the detection of Cu^2+^ both in solution and in the solid phase.

The syntheses based on the hydroxyphenyl derivatives of benzothiazole were reported as fluorescent probes **88a**–**c** for the detection of esterase in curing various diseases [60,66], and trace amounts of Hg^2+^ [67], Cu^2+^ and S^2–^ ions [68] were found (Figure 62). 

Nitrophenyl 2-(2-hydroxyphenyl)benzothiazole derivatives with –Ar–, –Ar–C=C– and –Ar–C≡C– linkers have been synthesized by the Suzuki, Heck, and Sonogashira reactions, respectively (Figure 63) [136]. The presence of the strong electron acceptor group 4-NO_2_C_6_H_4_ facilitates a charge transfer and affects the photophysical properties of the molecules. It also facilitates various intermolecular interactions. In the Heck reaction, the substrate was first acetylated with acetic anhydride, and the formed acetate was introduced to the reaction with (*E*)-4-nitrostyrene to obtain the acetate-protected product. Further deprotection under alkaline conditions gave the target product d-HBT-NO_2_. To investigate the fluorescent properties of the products, they were converted to the corresponding methoxy derivatives by the action of methyl iodide. Nitrophenyl 2-(2-anisyl)benzothiazole **89b** was found to be most promising for further investigation.

The Pd(PPh_3_)_4_-catalyzed Suzuki coupling of 2-(benzothiazol-2-yl)-5-bromophenol and commercially available carboxylic acids gave three positional isomers **90a**–**c** (Figure 64) [69]. The products showed strong emission in both solid and aggregated states and a low emission in solvents of different polarities.

Benzothiazole-2-carbaldehyde was used for the synthesis of new anti-HIV drug biotin-BMMP **91** [37]. First, the aldehyde was quantitatively reduced with NaBH_4_ to the corresponding alcohol, which was brominated with PBr_3_. The bromine atom in the formed 2-(bromomethyl)benzothiazole was replaced by pyrimidine thiol, as shown in Figure 65. Subsequent hydrazinolysis and the EDC-mediated conjugation of primary amine with biotine gave the target biotine-BMMP in 96% yield.

2-Acetylbenzothiazole is often used for the synthesis of hybrid molecules. Benzothiazoles containing thieno[2,3-*b*]pyridine moieties **92a** and **92b** were obtained in two steps: the bromination of 2-acetylbenzothiazole; and cyclization with mercaptonicotine nitrile (Figure 66) [137]. 

2-Acetylbenzothiazole has also been used for the synthesis of thiazole-, benzothiazole-, and benzofuran-containing molecules, as well as bis-benzothiazole derivatives. The main advantage of these reactions is their easy handling and cheap starting materials [105]. The transformations leading finally to benzothiazoles **93a**–**c** with ethylidenehydrazinyl linkers are shown in Figure 67. The components of condensation were prepared by the reaction of 2-acetylbenzothiazole with thiosemicarbazide or with bromine. The subsequent reactions of compounds **A** and **B** gave the target hybrid molecules. 

2-Bromacetyl benzothiazole reacts with mono- and bis-N-amino-2-mercaptotriazoles to give hybrid molecules **94a,b** and **95a**–**d** with one or two triazolothiadiazine moieties (Figure 68) [105].

In a similar way, 2-bromacetyl benzothiazole with bis(thiosemicarbazones) affords hybrid molecules 96a–c linked by the aliphatic spacer via phenoxy groups (Figure 69) [105].

Condensation of benzothiazole-2-carbohydrazide with 1*H*-indole-3-carbaldehydes gives rise to the formation of N-acylhydrazone derivatives **97a**–**e** possessing antitumor activity (Figure 70) [19]. The products are shown to exist as the *E*-diastereomers. The method is characterized by mild conditions, high yields, and easy handling.

Benzothiazole-2-carboxyhydrazide cyclizes with carbon disulfide in the presence of alkali to give the product containing a pharmacophore active oxadiazole motif [46]. Its aminomethylation with formaldehyde and primary or secondary amines allowed the authors to prepare a large series of benzothiazole-based oxadiazole Mannich bases, demonstrating its enhanced antidiabetic activity **98a**–**v** (Figure 71).

A series of benzothiazole-based condensed derivatives with 1,3,4-oxadiazole fragments **99a**–**j** with pronounced biological activity were synthesized via a multistep reaction sequence [23]. In the presence of 1-ethyl-3-(3-dimethylaminopropyl)carbodiimide (EDCI) and hydroxybenzotriazole (HOBt), benzothiazole-2-carboxylic acid reacts with 4-hydroxy-3,5-dimethoxybenzohydrazide to form hydrazide, which cyclizes via thionation with Lawesson’s reagent. Esterification of the product of cyclization and subsequent hydrazinolysis and cyclization with substituted benzoic acids afford new polyfunctional heterocycles **99a**–**j** (Figure 72).

The synthesis of highly sensitive probes for the detection of chemical warfare agents **100a,b** by the reaction of diethyl chlorophosphate with benzothiazole containing iminocoumarine residue in the C-2 position has been reported [138]. The target products were synthesized by the use of triethylamine, conc. hydrochloric acid, and organic Good’s buffers (Figure 73).

The functionalization of the phenylene fragment of benzothiazole is another possibility of modification. However, we were able to find only one example of such a transformation. A microwave-assisted regioselective three-component reaction of 2-methyl-5-aminobenzothiazole, aromatic aldehydes and 2-hydroxy-1,4-naphthoquinone in acetic acid afforded polycyclic condensed acridine derivatives **102a**–**h** [102]. The sequence of reactions included the Knoevenagel reaction, the intermolecular Michael addition with subsequent intramolecular nucleophilic cyclization, and the reactions of dehydration and oxidation. The MW-assisted [2+2+1] cyclization of acridinediones **101a**–**n** with aldehydes in the presence of ammonium acetate results in the oxazolole–thiazolole-condensed acridine ensembles **102a**–**h** (Figure 74). The proposed procedure is simple to perform, uses readily available reagents, provides selective modification of the acridine framework, and is characterized by a high efficiency of bond formation.

## 6. Conclusions

In summary, the versatile range of synthetic approaches to the C-2 derivatives of benzothiazole developed in the last five years is indicative of the relentless interest in this heterocycle, which is very promising from both a synthetic and biological point of view. In the present review, the methods of synthesis of the title compounds were divided into: (i) intra- and (ii) intermolecular assembling of the benzothiazole ring, (iii) the introduction of substituents at the 2-position, and (iv) the functionalization of the phenylene fragment. Among them, those including the thiazole ring closure and the modification of substituents at the C-2 position were dominant. Along with traditional multistep synthetic methods, new ecologically friendly atom economy one-pot procedures have been developed, which are the basis of modern organic synthesis. For the most interesting processes, only tentative mechanisms are given. Recent studies in this field have allowed the discovery of new C-2-substituted derivatives of benzothiazole and proven them to be good candidates for numerous drugs with various types of biological activity. Their pharmacological and biological activity strongly depend on the nature and position of the substituents, both in the benzene ring of the benzothiazole cycle and in the heterocycles formed by the functionalization of benzothiazole. The authors hope that this review will help the development of the targeted synthesis of benzothiazoles and their analogues.

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
