# Peer review of "Synthetic Approaches to Biologically Active C-2-Substituted Benzothiazoles"

_molecules, 2022, doi:10.3390/molecules27082598_

Round 1

Reviewer 1 Report

The paper, presented to me for review concerns chemistry of C-2 substituted benzothia-zoles. The research motivation, problem statement and research objective is not relate to each other.

It can be accepted for publication, but only after a major revision.

The text in the manuscript is inconsistent and it need to be re-written. Article has no good division into chapter. The idea is good. However, the manuscript no has well-formed conclusions. They are too general, not related to the topic. Detailed text analysis is not necessary at this stage. First, he has to be systematized and improved.

Author Response

(Only critical remarks of the Reviewer are included)

The research motivation, problem statement and research objective is not relate to each other.

It can be accepted for publication, but only after a major revision.

The text in the manuscript is inconsistent and it need to be re-written. Article has no good division into chapter. The idea is good. However, the manuscript no has well-formed conclusions. They are too general, not related to the topic. Detailed text analysis is not necessary at this stage. First, he has to be systematized and improved.

We tried to do our best in improving the manuscript. Thus, Abstract, Introduction, Conclusions, and the main body of the text are now substantially revised.

Reviewer 2 Report

This is a very comprehensive review that describes the synthetic approaches to the C-2 benzothiazole derivatives. In addition to the traditional methods that use transition metal catalysis, radical chemistry to synthesize benzothiazoles, this review also includes recent literatures that use atom economy and eco-friendly methods. Overall, this is a very comprehensive review article on benzothiazole derivatives synthesis that is worthy of publication in Molecules.

The manuscript in its current form is not recommended for publication in Molecules. The reaction schemes need to be addressed as they lack proper format. Chemdraw structures aren't consistent throughout the paper and so is the flow of the article. Overall, the article needs major revisions before it can be accepted for publication.

A lot of typo mistakes to be addressed.

The schemes and compound numbers are not uniform. Pay more attention to correct them.

In Scheme 1, show the generation of methyl radical from DTBP.

Mention the catalysts in scheme 3.

Mechanism shown in scheme 15 should be analyzed (for mistakes if any).

Mechanism in scheme 17, SH attack on iminium carbon is not shown. Include that structure to eliminate confusion.

This is a never-ending list. I have noted few of them. The authors should go through the manuscript carefully and I suggest to proof read the manuscript with language experts.

Author Response

(Only critical comments of the Reviewer are included)

The manuscript in its current form is not recommended for publication in Molecules. The reaction schemes need to be addressed as they lack proper format. Chemdraw structures aren't consistent throughout the paper and so is the flow of the article. Overall, the article needs major revisions before it can be accepted for publication. A lot of typo mistakes to be addressed.

We are sorry for not properly checking the text. This is explained (but by no means justified) by our desire to submit the review ASAP. In the revision, we tried to do our best to improve it.

The schemes and compound numbers are not uniform. Pay more attention to correct them. In Scheme 1, show the generation of methyl radical from DTBP.

Corrected.

Mention the catalysts in scheme 3.

Corrected.

Mechanism shown in scheme 15 should be analyzed (for mistakes if any).

The Reviewer is absolutely right. The mechanism given in the original paper raised many questions for us, too. Now, in the revision, we present virtually the same but more rational scheme, which, we hope, would not cause such questions.

Reviewer 3 Report

There are several recent reviews such as:

1- A Review on Anticancer Potentials of Benzothiazole Derivatives (2020). DOI: 10.2174/1389557519666190617153213

2- Biological Aspects of Emerging Benzothiazoles: A Short Review (2013). DOI:10.1155/2013/345198

3- A review on Indole and Benzothiazole derivatives its importance (2019). Journal of Drug Delivery and Therapeutics 9(1-s):505-509. DOI:10.22270/jddt.v9i1-s.2358

4- A Review of Environmental Occurrence, Fate, Exposure, and Toxicity of Benzothiazoles. Environ. Sci. Technol. 2018, 52, 9, 5007–5026

5- Advancement in Pharmacological Activities of Benzothiazole and its Derivatives: An Up to Date Review (2021). https://doi.org/10.2174/1389557520666200820133252

1- The authors don't show what is the importance of that review. That should be declared in the abstract. The aforemetioned reviews are some from other numerous previous and recent ones. 

2- The drawing in Schemes are not as good from bond lengths, angles and ring sizes.

3- The style of the review is not uniform. From example, line-space in page 2, last paragraph.

In my opinion, the review should be of keen redrawn and the abstract should be more declared

Author Response

(Only critical comments of the Reviewer are included)

There are several recent reviews such as:

  1. A Review on Anticancer Potentials of Benzothiazole Derivatives (2020).

DOI: 10.2174/1389557519666190617153213

  1. Biological Aspects of Emerging Benzothiazoles: A Short Review (2013). DOI:10.1155/2013/345198
  2. A review on Indole and Benzothiazole derivatives its importance (2019). Journal of Drug Delivery and Therapeutics 9(1-s):505-509. DOI:10.22270/jddt.v9i1-s.2358
  3. A Review of Environmental Occurrence, Fate, Exposure, and Toxicity of Benzothiazoles. Environ. Sci. Technol.2018, 52, 9, 5007–5026
  4. Advancement in Pharmacological Activities of Benzothiazole and its Derivatives: An Up to Date Review (2021). https://doi.org/10.2174/1389557520666200820133252

We thank you for the comment; four of five reviews are now included in the revision. The exception is review [2] because it is beyond the covered period (since 2016).

  1. The authors don't show what is the importance of that review. That should be declared in the abstract. The aforemetioned reviews are some from other numerous previous and recent ones. 

We declared it in the revised Abstract according to your recommendation.

  1. The drawing in Schemes are not as good from bond lengths, angles and ring sizes.

This is a technical comment; the drawings can be revised when/if the Editor specifies the structures.

  1. The style of the review is not uniform. From example, line-space in page 2, last paragraph.

Corrected.

 In my opinion, the review should be of keen redrawn and the abstract should be more declared

The Review is substantially redrawn. The abstract is also revised taking into account the form used in the reviews kindly recommended by you.

Round 2

Reviewer 1 Report

Improved manuscript is an already better written and prepared. Better to read and follow in compared to older version.

Comments to be considered, in order to further improve the manuscript quality:

  1. Please explicitly specify the novelty of your work. What progress against the most recent state-of-the-art similar studies was made?
  2. Avoid lumping references as in [1-7] and all other. Instead summarize the main contribution of each referenced paper in a separate sentence.

Reviewer 2 Report

I do not think authors have addressed all the comments. In my previous report I commented below points too.

Mechanism in scheme 17, SH attack on iminium carbon is not shown. Include that structure to eliminate confusion.

This is a never-ending list. I have noted few of them. The authors should go through the manuscript carefully and I suggest to proof read the manuscript with language experts.